# Strong ethanol- and frequency-dependent ecological interactions in a community of wine-fermenting yeasts

Simon Lax [1✉] & Jeff Gore [1✉]

Natural wine fermentation depends on a complex consortium of native microorganisms rather than inoculation of industrial yeast strains. While this diversity of yeasts can result in an increased repertoire of wine flavors and aromas, it can also result in the inhibition of *Saccharomyces cerevisiae*, which is uniquely able to complete fermentation. Understanding how yeast species interact with each other within the wine-fermenting community and disentangling ecological interactions from environmental impacts on growth rates, is key to developing synthetic communities that can provide the sensory benefits of natural fermentation while lowering the risk of stuck ferments. Here, we co-culture all pairwise combinations of five commonly isolated wine-fermenting yeasts and show that competitive outcomes are a strong function of ethanol concentration, with frequency-dependent bistable interactions common at low alcohol and an increasingly transitive competitive hierarchy developing as alcohol increases. We also show that pairwise outcomes are predictive of five-species community outcomes, and that frequency dependence in pairwise interactions propagates to alternative states in the full community, highlighting the importance of species abundance as well as composition. We also observe that monoculture growth rates are only weakly predictive of competitive success, highlighting the need to incorporate ecological interactions when designing synthetic fermenting communities.

[1] Physics of Living Systems, Department of Physics, MIT, 400 Technology Square, NE46-609, Cambridge, MA 02139, USA.  ✉email: simonlax@mit.edu; gore@mit.edu

Over the last decade, fermented foods have become an increasingly important model system within microbial community ecology. In addition to having direct economic importance, the microbial communities of fermented foods reproducibly assemble and have known culture conditions, allowing them to be broken apart and reassembled to directly test the nature of the interactions within the community. Many fermented foods, such as cheese rinds[1] or fermented meats and vegetables[2,3], are used as models for ecological succession, in which an initially bare substrate is colonized by an increasingly diverse and metabolically interconnected community of yeasts and bacteria[4]. Alcoholic fermentation, especially of wine grapes, presents an intriguing contrast, in which the buildup of ethanol as a byproduct of sugar consumption stresses the community to the point of eventual collapse. Understanding how interactions within the wine-fermenting community change as nutrients become depleted and alcohol concentration rises may provide general insights into how communities behave in deteriorating environments.

While the majority of commercially produced wine is fermented by inoculated industrial strains of *Saccharomyces cerevisiae*, spontaneous alcoholic fermentation of grapes is initiated by the complex microbial communities found on grape skins and on winery equipment. This native microbial community may include yeasts belonging to dozens of different genera, with abundances varying over more than five orders of magnitude[5]. While *Saccharomyces cerevisiae* is uniquely able to finish fermenting wine to dryness, due to its ability to withstand high ethanol concentrations, low nutrient availability, and anaerobic conditions[6], it is generally not abundant at the outset of spontaneous fermentations and represents a trivial fraction of most grape-associated microbial communities[5,7]. Instead, fermentation is initially dominated by weakly fermentive strains, such as *Metschnikowia pulcherrima* and *Hanseniaspora uvarum*, which are found in high abundance on grape skins[5]. As ethanol concentration rises, and these initially dominant species are no longer able to survive, the community becomes dominated by more strongly fermentive yeast strains such as *Lachancea thermotolerans* and *Torulaspora delbrueckii*, and eventually by *Saccharomyces cerevisiae*, which completes the fermentation process alone[5].

As *Saccharomyces cerevisiae* is both critical to a successful wine fermentation, and very rare at its outset, it raises the question of whether fermentation dynamics are governed primarily by a tradeoff between the initial abundance and the fermentive strength of the constituent yeast strains, or whether there are meaningful ecological interactions between the strains that change alongside the environmental conditions. It is well established that many non-*Saccharomyces* yeasts can withstand ethanol concentrations significantly higher than those at the point where they drop out of multispecies fermentations, implying that competition between strains plays an important role in fermentation dynamics[6,8]. Yeast strains within a fermentation have the ability to compete against each other through a variety of mechanisms, including competition over nutrients, secretion of toxins or inhibitory compounds, direct cell to cell contact, and quorum-sensing mediated modifications of metabolism[9]. Each of these mechanisms is in turn a function of the environment, and may be dependent on nutrient availability or alcohol concentration within the fermentation medium. Abiotic factors such as sulfate concentration or temperature have also been shown to modulate these interactions[10]. Numerous studies have characterized wine-fermenting communities over the course of experimental fermentations, helping to shed light on how the presence or absence of different species alters fermentation dynamics[5,6,8]. Still, it is difficult to infer interspecies interactions from transient population dynamics, and to disentangle the rapidly changing environmental conditions that occur during a fermentation from those interactions.

Understanding these ecological interactions has particular importance in the design of synthetic fermenting communities, which have the potential to provide some of the sensory benefits of spontaneous fermentations without the risk of stuck fermentations or undesirable spoilage organisms[11]. While they do not survive long into the fermentation, initially abundant species can play a profound role in shaping wine flavor and aroma, in both positive and negative ways. Many aromatic and flavor compounds, such as terpenes and thiols, are found in grapes as conjugated precursors that are odorless and flavorless until they are liberated by microbial glucosidases and β-lyases, respectively[12]. Unwanted microbial enzymes, like sulfate reductase, may also negatively impact wine quality[12]. While the enzymatic capabilities of different non-*Saccharomyces* yeast strains have been well studied, assembling a synthetic community is not as simple as mixing together several desirable strains. It is important to understand the ecological interactions within the community, as the abundance of early-successional species can influence the abundance of more fermentive yeasts, either through inhibition or facilitation[8,10]. Of particular concern is that *Saccharomyces* may become inhibited through ecological or environmental factors, resulting in a stuck fermentation that is difficult to reverse, and which can represent a large loss to producers.

Here, we directly measure all pairwise interactions within a five species consortium of commonly isolated wine yeasts. By employing a daily dilution protocol that holds ethanol concentration constant, we are able both to capture the equilibrium competitive outcomes within the community, and to observe how those outcomes change as a function of alcohol. We show that at low alcohol the interaction network is highly frequency dependent, but that it becomes increasingly transitive as alcohol concentration increases, with *Saccharomyces* becoming the competitive dominant. By varying the initial abundances of the species within our consortium, we show that this frequency dependence in pairwise interactions propagates to the full community, highlighting the importance of species abundances at the outset of fermentation. Finally, by running experimental ferments with the same five species, we demonstrate that an understanding of how ecological interactions change with ethanol concentration can contextualize the success or failure of a multispecies fermentation.

## Results

Our initial aim in this study was to directly measure the competitive outcomes within a wine-fermenting community as a function of ethanol concentration. We employed a consortium of five commonly isolated wine-fermenting yeast species, which have uniquely identifiable colony morphologies (Fig. 1a): *Candida railensis* (CR), *Hanseniaspora uvarum* (HU), *Metschnikowia pulcherrima* (MP), *Saccharomyces cerevisiae* (SC), and *Tourlaspora delbrueckii* (TD). To disentangle competitive outcomes from the environmental impacts of rising ethanol, we employed competitive assays which held ethanol concentration constant through daily dilutions into fresh juice with a set ethanol concentration. In essence, this allowed us to freeze the environmental conditions at key points in the fermentation to understand whether the competitive dynamics change as a function of rising ethanol, which is difficult to infer from the transient dynamics of an experimental fermentation. While other environmental factors beyond ethanol concentration change over the course of a fermentation, in particular the depletion of carbon, nitrogen, and other key nutrients, we focused on the early stages of the

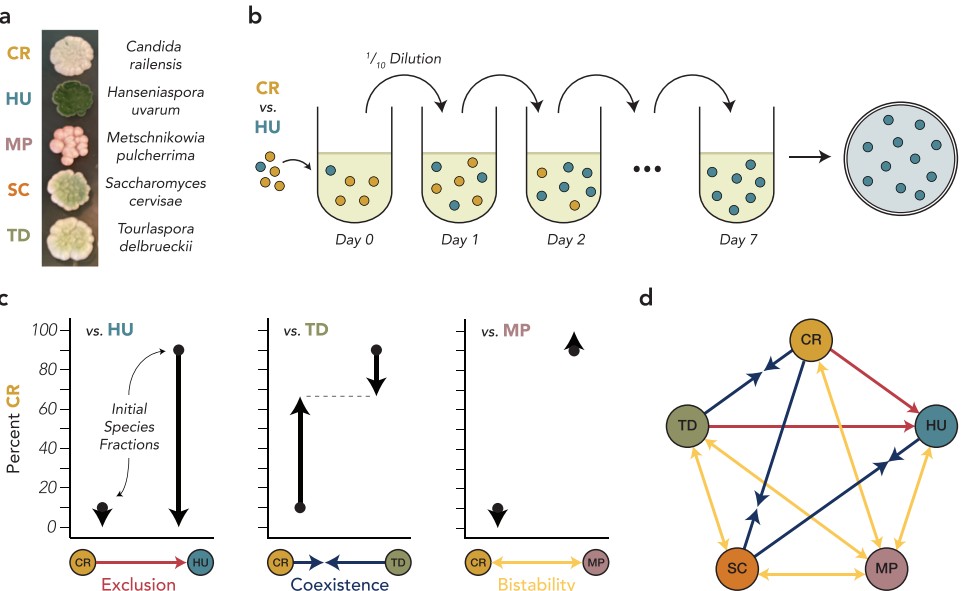

**Fig. 1 Pairwise competitive outcomes within a consortium of wine-fermenting yeasts in the absence of ethanol reveal a mixture of coexistence, exclusion, and bistability. a** The five species of wine-fermenting yeasts in this study have colony morphologies that can be visually differentiated when plated on Wallerstein Laboratory (WL) nutrient agar. **b** To determine the equilibrium competitive outcomes between pairs of yeasts in the absence of ethanol accumulation, we co-cultured species with daily dilutions into fresh grape juice. The final species fraction was determined by plating the community onto WL agar. **c** There are three possible competitive outcomes for each pair of species, which were determined by starting the co-cultures at two initial species fractions. One species may exclude the other regardless of the initial fraction (left), the two species my coexist at a fraction reached by both initial communities (center), or the winner of the competition may be determined by the initial species fraction (right). **d** The full interaction network reveals that the majority of the competitive outcomes between these species are dependent on the initial species fraction, while only two pairs of species exhibit competitive exclusion.

fermentation trajectory (less than 5% ethanol by volume), when cells are most actively growing and before most resources become depleted. We also found that removing up to 75% of the carbon or nitrogen in a synthetic grape juice-like media had negligible impacts on the growth of each of our five species (Supplementary Fig. 1).

We began by characterizing the pairwise competitive relationships between the species in the absence of ethanol accumulation (Fig. 1b), coculturing our species in sterilized Gewürztraminer juice. The cocultures were initiated at two initial species fractions, where one of the two species was initially dominant (99% of the inoculated cells). At the end of the 7-day cycle, the community was plated to determine the equilibrium community composition. We observed three qualitatively different possible outcomes in our pairwise competitive assays (Fig. 1c): one species may outcompete the other (i.e. CR is excluded by HU), the two species may coexist at a stable fraction (i.e. CR coexists with TD at a 2:1 ratio), or the outcome may be bistable, where the winner of the competition depends on the initial species fraction (i.e. CR excludes MP when initially dominant but MP excludes CR when initially dominant). Each pairwise competition was replicated at least three times (see Supplementary Fig. 2), and we built a consensus interaction network by averaging the replicated outcomes (Fig. 1d). We found that in the absence of ethanol accumulation, the network was dominated by bistable outcomes, driven largely by MP, which had a bistable relationship with every other species. Interestingly, *Saccharomyces* was never a competitive dominant in the absence of ethanol accumulation: it coexisted with CR and HU and had a bistable relationship with MP and TD, implying that high initial abundances of either of those latter two species may inhibit *Saccharomyces* growth early in a fermentation.

The ubiquity of frequency-dependent pairwise interactions implies that it may be difficult to predict the behavior of the full

five species community, and that alternative equilibrium states may exist that are conditional on the initial abundances. To test this, we ran competition assays for mixtures of all five species, always starting with one initially dominant species (96% of the inoculated cells) and four low abundance species (1% of the inoculated cells each). As in the pairwise assays, the community was diluted into fresh juice every 24 h to keep ethanol from accumulating, and after the final dilution cycle the community was plated to determine the equilibrium community structure (Fig. 2a). Each initial condition was replicated at least 10 times, and the results were surprisingly divergent even across biological and technical replicates (Fig. 2b). Still, some striking differences were evident based on the initial condition. When communities were initiated with CR, HU, or SC as the dominant species, they uniformly converged to a mixture of HU and SC with HU as the dominant species. When MP was the initially dominant species, it consistently excluded HU, leading to MP or SC dominance, or some mixture of the two that is likely transient given their bistable relationship. Communities in which TD was initially dominant had the widest range of outcomes, with every species reaching greater than 10% final abundance in at least one replicate. Notably, CR was able to survive within the community at low initial abundance when TD was the initially dominant species, and not when it was initially high abundance, demonstrating how community structure at the outset can influence successional trajectories in non-intuitive ways.

To formalize the complex set of competitive outcomes, we classified them into four canonical structures that represent four possible stable points of attraction on the pairwise interaction network (Fig. 2c). Each replicate was assigned to the structure it most closely resembled (which occasionally meant that replicates which resulted in exclusion were assigned to states with expected coexistence, or vice versa), and the probability of the community reaching a given structure was a function of the initial

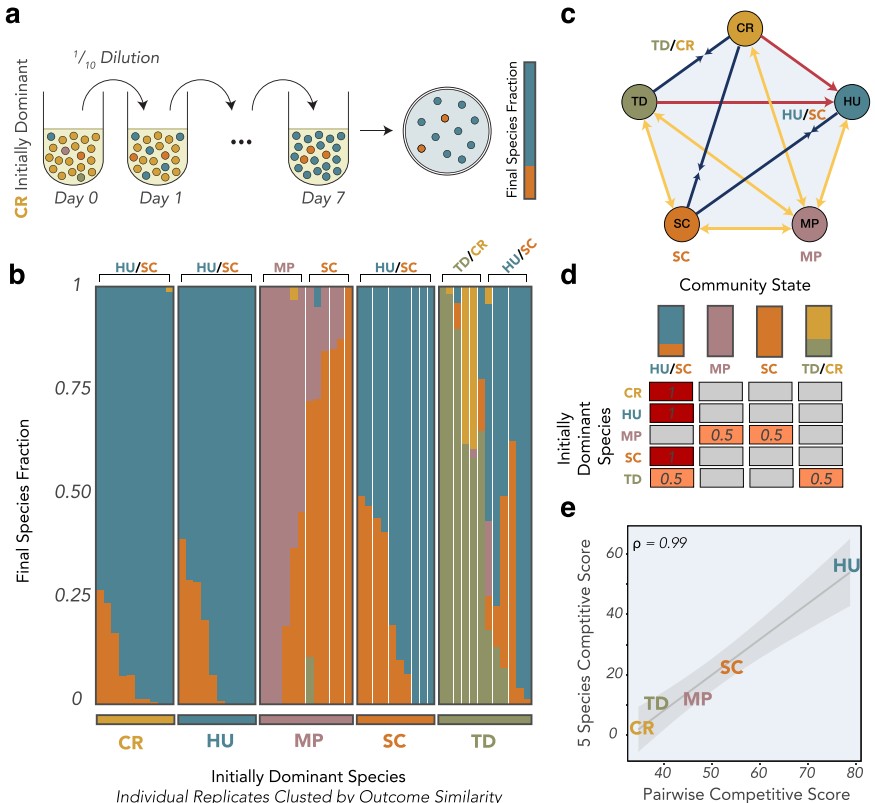

**Fig. 2 Density-dependence in pairwise interactions propagates to alternative states in the five-species community. a** To understand how initial species fractions alter the competitive outcomes in a community of all five species, we initiated co-cultures in which one species was initially dominant (96% of cells) and the other four were present at much lower initial abundance (1% of cells each). As in the pairwise competitions, the community was diluted daily into fresh juice to keep ethanol from accumulating, and the final species fraction was determined by plating onto WL agar. **b** Final species fractions for the five-species communities, clustered by the initially dominant species. Each initial community was replicated at least 10 times, and each stacked bar represents an individual replicate. **c** The final 5-species community states can be mapped onto the pairwise interaction network, with four centers of attraction on the network dependent on the initial community state. **d** The probability of the community reaching each of the four centers of attraction is dependent on the initially dominant species. **e** Competitive success in pairs and in the 5-species communities, as quantified by our competitive score metrics, are highly correlated.

abundances (Fig. 2d). All the CR-, HU- and SC-dominated replicates converged to an HU-dominated coexistence with SC. MP-dominated communities were equally likely to wind up closest to MP dominance or SC dominance, while TD-dominated communities were equally likely to wind up closest to HU/SC coexistence or to coexistence between TD and CR. Although somewhat noisy, these alternative outcomes demonstrate how bistable pairwise relationships, driven here by MP and TD, can lead to frequency dependence in the fermentation successional trajectory, implying that the initial abundances of wine-fermenting community members may be just as important as the species composition within the community in determining fermentation outcomes.

To quantify the competitive success of each species, we assigned competitive scores as a function of both the pairwise and 5-species coculture results. For the pairwise competitions, we assigned scores on the following basis, which were averaged across all four pairs that included the given species: 100 points for excluding the other species, 0 points for being excluded by the other species, 50 points for a bistable outcome, and $X$ points for coexisting with the other species at percentage $X$. For the 5-species competitive score, we took the simple average of the percent abundance the species reached across all 5-species community replicates. Interestingly, these two scores were almost perfectly correlated (Pearson correlation = 0.99, $p = 0.002$) (Fig. 2e), highlighting how an understanding of pairwise

ecological interactions can provide strong insights into the behavior of artificially assembled communities, even in the face of frequency-dependent alternative states.

We next explored how changing the alcohol concentration in the coculture experiments affected the competitive outcomes. By adding a set ethanol quantity to the grape juice at each dilution point, we held ethanol constant at 6 concentrations between 0 and 5% alcohol by volume (ABV). We found that the interaction networks changed dramatically even between small increments of ethanol concentration (Fig. 3a), with the networks becoming increasingly dominated by competitive exclusion. While competitive outcomes in the absence of ethanol accumulation were highly frequency dependent, competitive outcomes at 4 and 5% ABV were almost completely transitive, with SC winning all pairwise competitions, TD and HU winning all other competitions but bistable between themselves, MP excluded by all species except CR, and CR excluded by all other species. We also ran 5-species competition experiments as in Fig. 2 at 2 and 5% ABV. At 2ABV (Fig. 3b), we found that almost all initial communities gravitated towards coexistence between HU and SC, with only a small number of initially TD-dominated communities leading to TD dominance and one initially MP-dominated community resulting in an HU-excluded SC-MP coexistence that would likely have reached SC dominance with more time. The results at 5% ABV (Fig. 3c) were more surprising, given that the pairwise interaction network suggests that SC should exclude all other

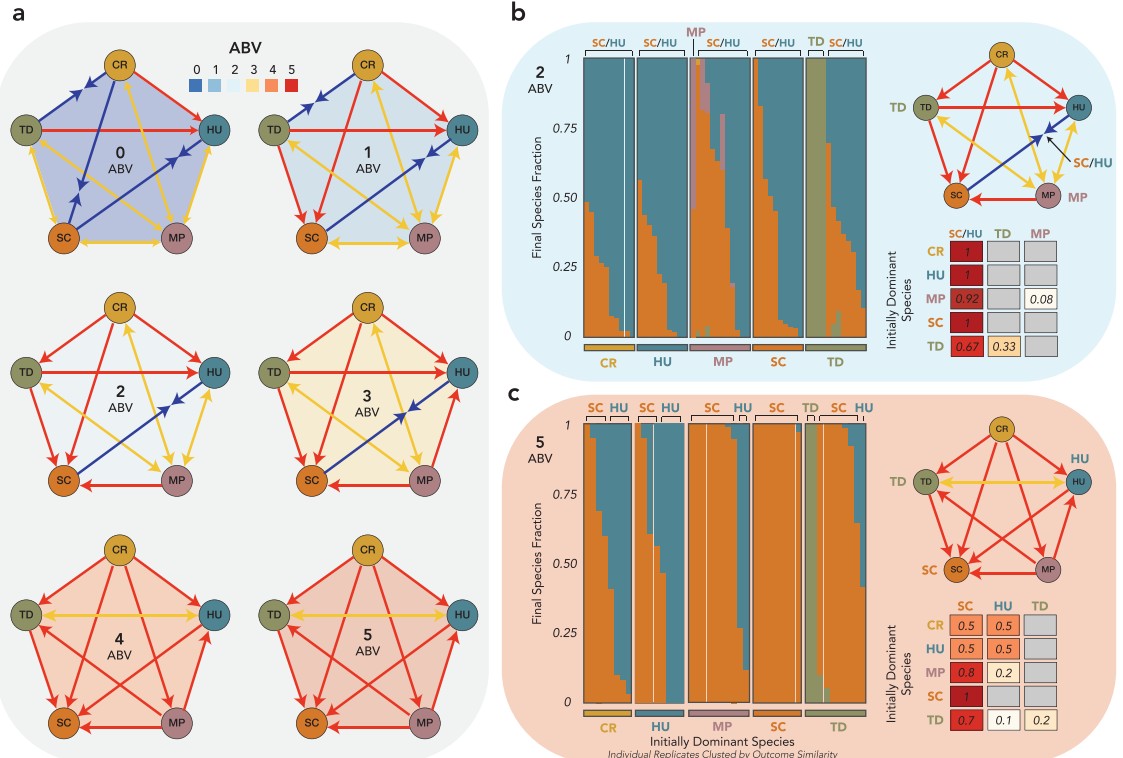

**Fig. 3 Interaction networks are altered by increasing ethanol concentration, leading to alternative states in the five-species community. a** Pairwise interaction networks as in Fig. 1D for 6 ethanol concentrations between 0 and 5% ethanol (ABV). **b** Five-species coculture results as in Fig. 2 for communities held at 2ABV. **c** Five-species coculture results as in Fig. 2 for communities held at 5ABV.

species. While most communities in which MP, SC, or TD was initially dominant reached SC dominance, communities in which CR or HU was initially dominant often included a large proportion of HU, with half of the communities closer to HU dominance than to SC dominance. Taken together, these results demonstrate how the disappearance of most bistable relationships with rising ethanol clears the way for *Saccharomyces* to dominate the fermentation community, even at low initial abundance.

To gain insight into the changing competitive outcomes, we measured the growth rate of each of our species as a function of ethanol concentration, using a time-to-threshold approach that implicitly incorporates lag time (see methods) (Fig. 4a). The growth rates of all species declined as ethanol concentration increased, although notably less so for SC. While only the fourth fastest grower at 0% ABV, SC became the fastest grower by 4% ABV, which is notably the concentration at which it first wins all pairwise competitions. Except for CR at 5% ABV, which did not have a measurable growth rate, all species had growth rates high enough to survive the imposed dilution factor at each ethanol concentration in this study, highlighting the importance of incorporating ecological interactions when predicating which species will survive within the community.

To better quantify how changing ethanol concentration affects the competitive outcomes between the 5 species, and how these outcomes are linked to growth rates, we assigned competitive scores to each species at each ABV, as in Fig. 2e. The rank order of the pairwise competitive scores was notably consistent, with only two changes in rank as ethanol increased: MP was surpassed by TD between 1 and 2% ABV and HU was surpassed by SC at 4% ABV (Fig. 4b). Despite the relatively consistent rank orders, the variance between scores increased as a function of ethanol, with the range in scores doubling from a ~50 point gap between the best and worst competitors at 0% ABV to a full 100 point gap

at 5% ABV. We found that both pairwise and 5-species competitive scores were usually positively related to growth rates (Fig. 4c, all statistics provided in Supplementary Table 1), except for the 5-species scores at intermediate ABV, although this relationship became stronger as ethanol concentration increased. For both score types, carrying capacity (measured as the optical density of the monoculture) was unrelated to competitive score at low ABV but became increasingly positively related at higher ethanol concentrations (Fig. 4d). As growth rates decline with higher ethanol, driven in large part by longer lag times, this increasingly positive relationship between carrying capacity and competitive ability may be driven by the benefit conferred by having a higher number of cells transferred at each dilution cycle. Pairwise competitive ability was related to competitive outcomes in the 5 species communities (Fig. 4e), and as previously noted notably had a very strong correlation in the absence of ethanol. While the frequency-dependent interactions within this consortium complicate our ability to predict the behavior of the full community from pairwise interactions, this strong correlation highlights how the competitive strength of individual strains collectively govern the multispecies fermentation dynamics. It's important to note that these correlations between competitive score and monoculture growth characteristics, although almost uniformly positive, rarely rise to the level of statistical significance (see Supplementary Table 1). While they offer some insight into why individual species are favored in a multi-species fermentation, they also highlight how they are not sufficient to predict fermentation dynamics without incorporating ecological interactions.

To better understand whether the competitive results we observed in Gewürztraminer are generalizable to other juice types, we replicated a subset of our experiments in Merlot juice. As in Fig. 3, we mapped the pairwise interaction networks at

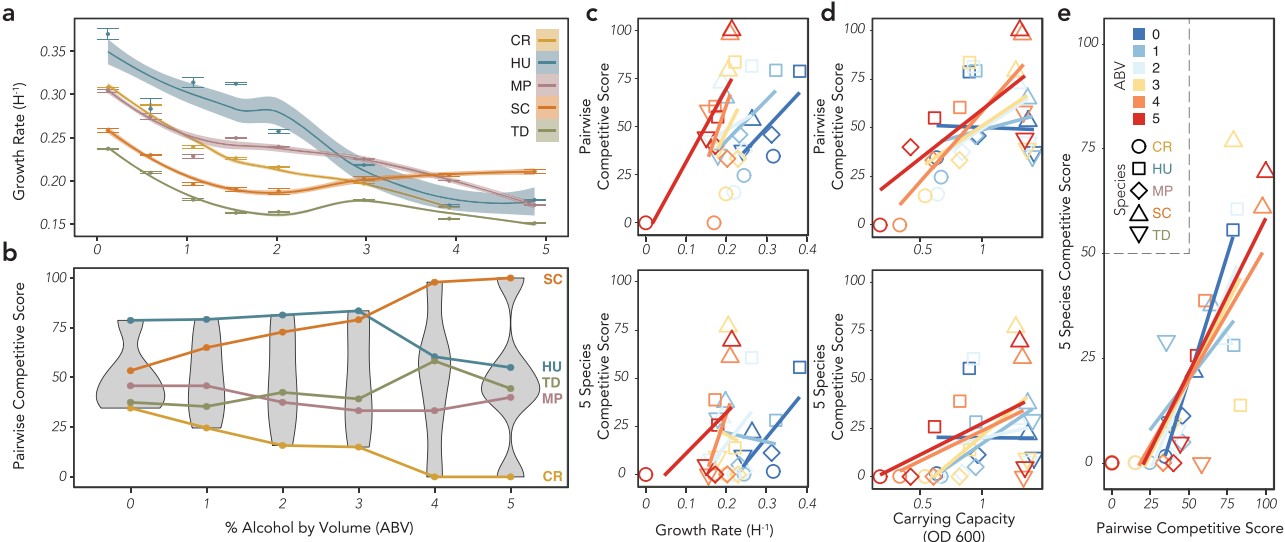

**Fig. 4 Competitive success in 2- and 5-species communities varies by alcohol concentration and is linked to monoculture growth rates and carrying capacities. a** Growth rates for the five yeast species as a function of ethanol concentration. Growth rate is measured as time to a threshold OD and implicitly incorporates lag time (see Methods). Error bars indicate the standard error of the mean for each measurement, and the trendlines are smoothed regressions with confidence intervals based on all growth rate measurements. **b** Pairwise competitive score ranks as a function of ethanol concentration. While the ranks are largely consistent across ethanol concentrations, SC only becomes the competitive dominant above 4% ethanol. Pairwise and 5-species competitive scores are almost always positively correlated to growth rate (**c**), correlated to carrying capacity at higher ABVs (**d**), and highly correlated to each other regardless of ABV (**e**).

varying ethanol concentration (Fig. 5a) and found that competitive exclusion was considerably more common than in Gewürztraminer, especially at lower alcohol concentrations. In Merlot, SC always excluded MP and TD, in contrast to their bistable interaction in Gewürztraminer. Additionally, HU excluded SC at all ethanol concentrations below 4% ABV in Merlot, in contrast to Gewürztraminer where HU was never found to exclude SC. We also assessed the 5-species dynamics, running competitive assays at 0 (Fig. 5b), 2 (Fig. 5c) and 4% ABV (Fig. 5d). Surprisingly, across almost all conditions (ABV and initially dominant species), the community converged to heavily HU-dominated coexistence with SC. The one exception was MP-dominated communities at 0% ABV, where MP excluded HU allowing SC to dominate the community. Interestingly, while individual competitive outcomes often differed between the two juice types, the aggregated pairwise competitive scores of each species as a function of ethanol were highly correlated (Pearson correlation = 0.92, $p = 3.6 \times 10^{-13}$) (Fig. 5e), although CR and TD were notably more successful in Gewürztraminer while HU was notably more successful in Merlot. In a comparison of the 5-species competitive scores (Fig. 5f), HU was a substantially stronger competitor in Merlot regardless of ethanol concentration, while SC was a stronger competitor in Gewürztraminer at higher ABV, highlighting how the ability of other strains to suppress *Saccharomyces* is a function of the environmental conditions as well as the composition and abundance of the community members.

Having characterized the equilibrium dynamics of our community when held at a constant ethanol concentration, we then turned to the dynamics of experimental fermentations, in which juice was left to ferment uninterrupted by any experimental interventions. We inoculated both Gewürztraminer and Merlot juice with monocultures of each of our species (at an initial density of $10^6$ cells per mL) and found that they varied widely in their fermentation capacities, with only SC able to ferment either of the juices to dryness (the absence of residual sugar) (Fig. 6a). Our Gewürztraminer and Merlot juices differed substantially in

their initial sugar concentration, with the Merlot's 25.5° brix (255 g/L of sugar) leading to a potential alcohol concentration of 15.5% ABV and the Gewürztraminer's 21.5° brix (215 g/L of sugar) leading to a potential 12.5% ABV. Both MP and CR struggled early in the fermentation, ceasing to ferment past a drop of ~ 5° brix. HU ceased fermenting after a drop of 8-9° brix and TD ceased fermenting after a drop of ~15° brix. While these strains differ in their fermentive strength, all can withstand ethanol concentrations in the range of the experimental conditions, highlighting how interspecies interactions are critical to predicting the successional trajectory of the community.

We ran mixed-starter fermentations in both juice types under three different experimental conditions, in which all five species were included but at different inoculation densities (Fig. 6b). To reflect its normally low abundance at the outset of natural fermentations, SC was always kept a low inoculation density of $10^4$ cells/mL. In the first condition (structure I), all four other strains were inoculated at the same density of $10^6$ cells/mL. In the second and third conditions, we chose a single species to initially dominate the community, with a starting density of $4 \times 10^6$ cells/mL and all other strains inoculated at $10^4$ cells/mL. We focused on MP- (structure II) and TD- (structure III) dominated communities, because both were found to have bistable relationships with SC in low-alcohol conditions, implying that they may be able to stall the fermentation by inhibiting growth of SC, the only species capable of finishing the fermentation.

In Gewürztraminer (Fig. 6c), both community structures I and II finished fermenting at the same time on day 22, with the evenly split structure I fermenting slightly faster at the outset but the MP-dominated structure II catching up in the later stages. Structure I was characterized by a rapid transition to SC/TD coexistence until day 11, when SC became the only surviving species. In contrast, TD was completely absent within the MP-dominated structure II, where SC became the dominant species at the outset and was the last surviving species by day 5. In the TD-dominated structure III, TD was the only species detected until day 11, when a small minority of SC cells first appeared. SC then rapidly took over and was the only

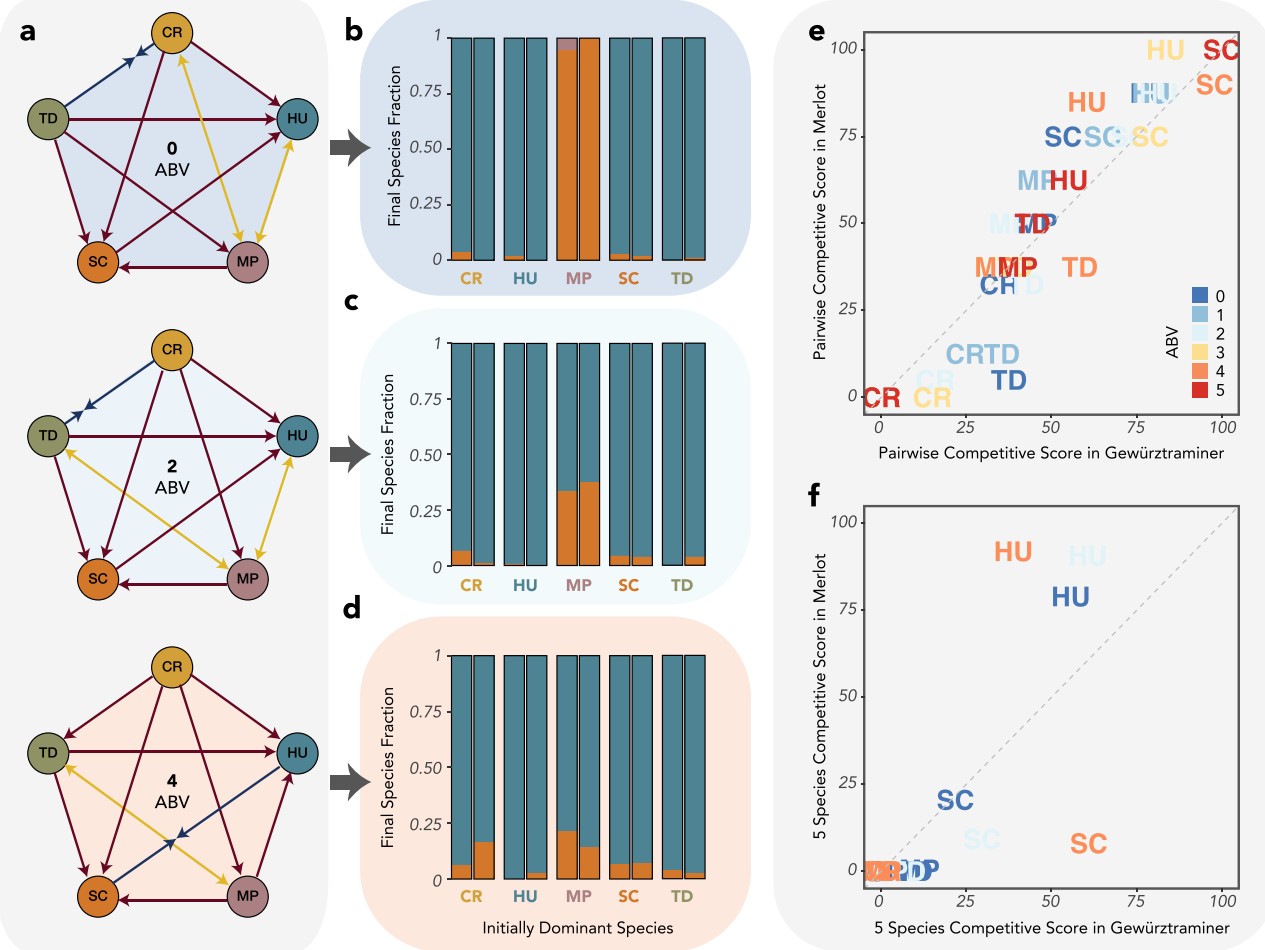

**Fig. 5 Competitive outcomes are influenced by juice type, although competitive scores remain highly correlated. a** Pairwise interaction networks in Merlot juice at 3 ethanol concentrations, as in Fig. 3a. **b–d** 5 species competitive outcomes as a function of the initially dominant species at 0 (**b**), 2 (**c**), and 4 (**d**) ABV. **e** Pairwise competitive scores in the two grape juice types are highly correlated (Pearson correlation = 0.92, $p = 3.6 \times 10^{-13}$). **f** Comparison of 5-species competitive scores in the two grape juice types. Note that SC and HU are the only species with a non-zero 5-species competitive score in Merlot.

detected species at the following sampling on day 13, but this initial TD dominance led to a longer fermentation that finished 6 days after the other two structures. The effect of the initial community structure on fermentation efficiency was even more evident in Merlot (Fig. 6d), where the MP-dominated structure II finished fermenting by day 30 while the other two structures had still not completed fermenting by day 45. In that structure, MP rapidly cleared the way for SC dominance by excluding TD, while in the other structures TD remained a significant community member until day 18. This benefit conferred to SC by initial MP dominance is somewhat counter-intuitive, as MP, SC, and TD all had bistable relationships with each other in the low-ABV cocultures. While the MP-SC bistability only persists until 2% ABV, the MP-TD bistability persists until 4% ABV, apparently giving sufficient time for MP to exclude TD early in the fermentation but not SC. Taken together, our results demonstrate that an understanding of how pairwise interactions change over the course of a fermentation can contextualize the success or failure of artificially assembled fermenting communities.

## Discussion

In this study, we directly map all the pairwise competitive relationships in a wine-fermenting yeast consortium as a function of ethanol concentration and contextualize these results in terms of the monoculture growth rates of each species and the competitive

results within the full five species community. While numerous studies have studied the behavior of each of these species in experimental fermentations, ours is the first study to our knowledge to explicitly measure how the competitive relationships between strains change over the course of the fermentation by holding the ethanol concentration constant and determining the equilibrium competitive outcomes. This bottom-up approach allows us unique insights into how interspecies relationships in the community are altered as the environment deteriorates.

At low alcohol, we show that competitive relationships within the wine fermenting community are highly frequency dependent. Importantly, this frequency dependence is not driven by Allee effects, a reduction in growth rate at low population density which could lead the population of the species with the low initial density to collapse[13]. Allee effects were not observed in the monoculture growth rate experiments for any of the five species and would be unlikely to play an important role in wine fermentation for several reasons. First, the primary sugars in ripened grapes are glucose and fructose, which do not allow for cooperative metabolism in the manner of sucrose, which must be hydrolyzed externally and can therefore benefit neighboring cells. Second, ethanol cannot be degraded or otherwise neutralized in the manner of other toxins such as antibiotics, foreclosing the possibility of cross-protection between cells. Ethanol inhibits cell growth and viability primarily by increasing the porosity of the yeast cell membrane, destabilizing the cells' electrochemical

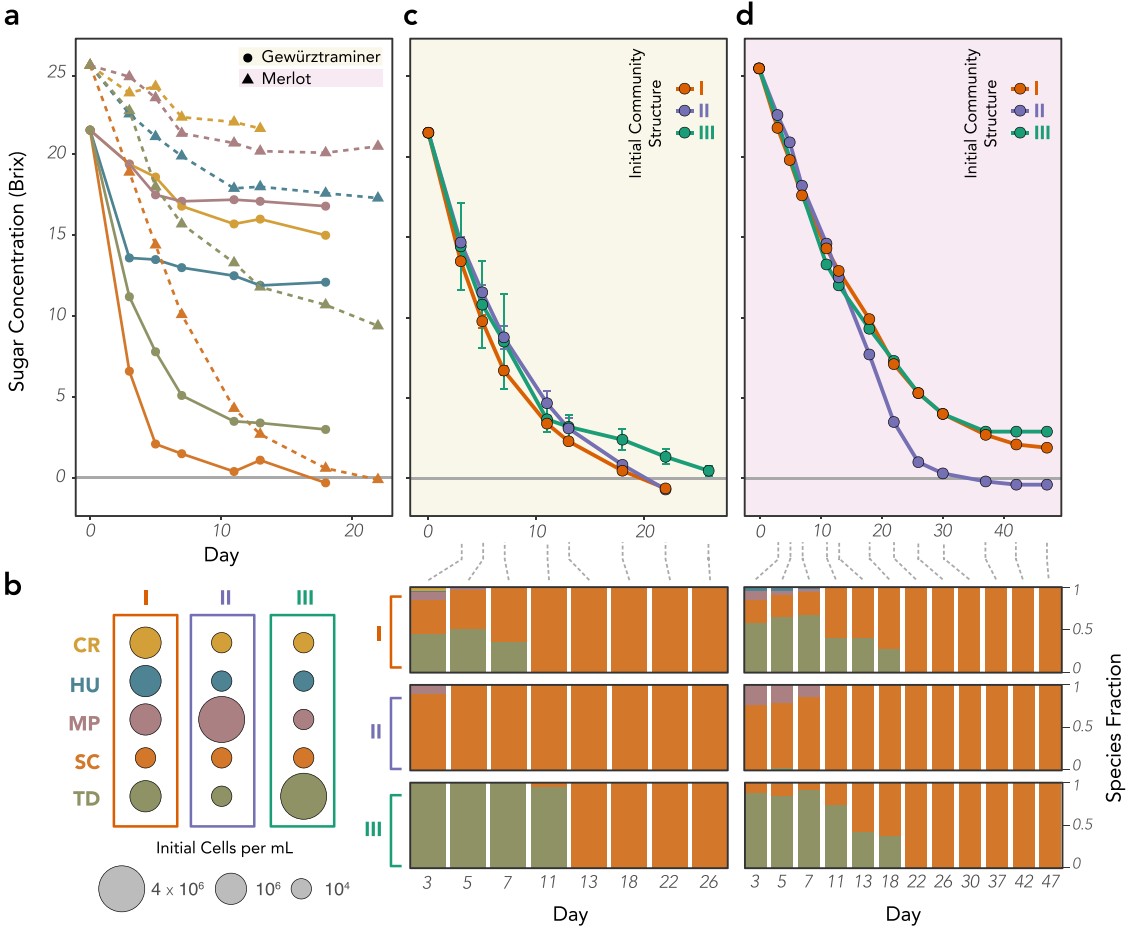

**Fig. 6 Equilibrium competitive outcomes provide insight into the dynamics of wine fermentation. a** Monoculture fermentation curves for each of the five yeast species in Gewürztraminer (solid lines) and Merlot (dashed lines). Only SC is capable of fermenting either juice type to dryness. **b** Fermentations were initiated at three initial species compositions (community structures) which reflect the alternative states observed in the equilibrium 5-species competitions. SC abundance was kept constant, with the majority of the community comprised of an even mix of the four other species (I) or dominated by MP (II) or TD (III). **c, d** The initial community structure impacted both the kinetics (top) and the successional trajectory (bottom) of the fermentation in Gewürztraminer (**c**) and Merlot (**d**). Error bars in (**c**) represent standard error of the mean.

gradients, and necessitating an 'every cell for themself' response, which relies on synthesizing new fatty acids for the cellular membrane[14]. While we treat ethanol concertation in this study as an environmental factor modulating the ecological interactions between species, it is important to note the production of ethanol is itself a competitive mechanism favoring more alcohol-tolerant species. This is especially true for *Saccharomyces*, which ferments even in the presence of oxygen when respiration would be more energetically favorable (the Crabtree effect)[15] and thereby increases its fitness through niche construction[16].

In the absence of Allee effects, this frequency dependence presumably stems from both species experiencing stronger interspecific (between-species) competition than intraspecific (within-species) competition, which may be driven by numerous factors. While we do not explicitly focus on the mechanisms behind the competitive relationships within our community, varying amounts are known about the metabolism and antimicrobial activities of each of our five species. MP, which at low alcohol has a bistable relationship with every other species in the consortium, has been well studied for its production of pulcherrimin, an insoluble red pigment with anti-fungal activity that acts by depleting iron within the growth medium and suppressing the growth of microorganism that require iron for their development. Pulcherrimin is known to suppress the growth of both *Candida* and *Hanseniaspora*[17], suggesting this may be a primary factor in those bistable relationships. Pulcherrimin is

not known to strongly suppress *Saccharomyces* growth[18], and there is even evidence that *Metchnikowia* can facilitate the growth of *Saccharomyces* by releasing amino acids through its extracellular hydrolytic enzymes, which can serve as nutrients for SC later in the fermentation when more accessible forms of nitrogen become depleted[19]. This may help contextualize why initially MP-dominant communities were the quickest to finish fermenting in our experiments. We also note that the effects of toxin-mediated competition may be particularly pronounced in our experiments which are characterized by high population densities in small volumes.

Of our four non-*Saccharomyces* species, *Torulaspora delbrueckii* has been the most well studied in multi-starter fermentations, due to its ability to decrease volatile acidity and increase the presence of fruit flavor and aroma[20], as well as for its high ethanol tolerance. There is some evidence that direct cell to cell contact can lead to death of TD by SC in cocultures, and TD grown in cell-free SC supernatants have also been found to be strongly inhibited[21]. While less is known about TD's ability to hinder SC growth, there are some TD strains which secrete a killer toxin (Kbarr-1) that has broad antifungal activity, including against *Saccharomyces*[22].

It's important to note that prior work on these species may provide only limited insights into our specific consortium due to the vast strain-level diversity that exists within wine yeasts. The killer phenotype observed in some *Saccharomyces* and *Torulaspora* strains

is a clear example of the importance of strain-level assessment, and the hundreds of different commercially available *Saccharomyces cerevisiae* strains speak to the profound differences in fermentation outcomes that can occur even with closely related strains. Additionally, while we focus primally on ethanol concentration in this study, there are numerous other environmental parameters, many left to the discretion of the winemaker, which can influence the interactions between wine-fermenting species. For example, the temperature of the fermentation can play a major role in the persistence of non-*Saccharomyces* yeasts within the community, with lower temperatures (10–15 °C) favoring *Candida* and *Hanseniaspora* persistence[23], and higher temperatures (25–30 °C) leading to quicker dominance by *Saccharomyces*[24]. The addition of sulfate at the outset of fermentation is also used to hasten *Saccharomyces* dominance by selectively inhibiting other yeast strains, including *Hanseniaspora* and *Torulaspora*[25]. The pH of the grape must, the osmotic pressure, and the availability of oxygen or nutrients also impact the dynamics of a fermentation[26,27], highlighting how the results of any one study can be difficult to generalize to other fermentation conditions. Because of the critical importance of strain-level diversity and fermentation conditions, we do not intend for the competitive outcomes we observe to be generalizable beyond the specific conditions of this study. Indeed, even changing the type of grape juice while holding all other experimental conditions constant led to meaningful changes in the interaction network. Instead, we intend this work to be a proof of principle that ethanol concentration is sufficient to modulate competitive relationships between yeasts in a fermentation, and that understanding these interactions and how they change over the course of a fermentation is critical when assembling synthetic wine-fermenting communities.

The equilibrium competitive results observed in the daily dilution experiments provided important insights to the dynamics of the experimental fermentations, particularly MP's ability to expedite the fermentation by clearing away SC's competition and TD's ability to stall the fermentation by suppressing SC at low alcohol. There was, however, one major incongruity: HU's almost complete absence in the experimental ferments despite its competitive strength in the in the cocultures. While the use of our daily dilution protocol allowed us to hold environmental conditions constant and measure the equilibrium competitive results, it does differentiate the environmental conditions from those of a traditional fermentation. Of particular relevance is the imposition of a constant death rate due to the daily dilutions, which in our case was equivalent to a 90% mortality rate imposed every 24 h. While each of our strains had a high enough growth rate at each ethanol concentration to withstand this death rate (except for CR at 5% ABV), this imposed morality can nonetheless alter the competitive dynamics, conferring favor to faster-growing species[28]. As HU is the fastest growing species in the community, up to 4% ABV, this may help explain its dominance in the coculture experiments compared to the uninterrupted ferments.

In conclusion, we show that ecological interactions play a dominant role in shaping wine-fermenting yeast communities, with monoculture growth rates and carrying capacities only weekly associated to competitive outcomes. We show that these competitive outcomes are a strong function of ethanol, with highly frequency-dependent interactions at low alcohol shifting to a *Saccharomyces*-dominated transitive competitive hierarchy at higher alcohol. Further, we show that these frequency dependent interactions at low ethanol can lead to alternative outcomes in the full community, highlighting how species abundance as well as species composition plays an important role in fermentation dynamics.

## Methods

**Species and medium**. We used a consortium of five yeast strains throughout this study. Two species, *Metschnikowia pulcherrima*

(StrainID n04-201) and *Tourlaspora delbrueckii* (StrainID n69-34), were obtained from the Phaff Yeast Culture Collection at UC Davis. The three other species, *Candida railensis*, *Hanseniaspora uvarum*, and *Saccharomyces cerevisiae*, were isolated from spontaneous ferments of store-bought table grapes, and identified through sequencing of the ITS1 marker region. To assess the impact of carbon and nitrogen availability on the growth rates of our five species, we used a synthetic grape juice-like medium adapted from Bely et al.[29]. At full carbon and nitrogen concentration, the medium contained 200 g/L sugars (100 g/L glucose and 100 g/L fructose) and 300 mg/L assimilable nitrogen (460 mg/L $NH_4Cl$ and 180 mg/L amino acids), with the pH adjusted to 3.5. All coculture experiments took place in sterilized grape juice, either in Gewürztraminer sourced from the Willamette Valley in Oregon (21.5 brix, pH 3.3) or Merlot sourced from the Walla Walla Valley in Washington (25.5 brix, pH 3.9). Juice was sterilized by autoclaving at 110 °C for 10 mins, and was subsequently plated to ensure its sterility.

**Coculture Experiments**. Frozen stocks of the competing species were streaked out on yeast-extract peptone dextrose agar plates and left to grow at room temperature for ~48H. A single colony was then picked and transferred to a 50-mL centrifuge tube containing 5 mL of sterilized juice, adjusted to the ethanol concentration of the future competition medium. Monocultures were then left to grow at room temperature for ~48H, before being standardized to a strain-dependent OD that corrected for OD to CFU variance between strains (0.1 for SC, 0.085 for TD, 0.075 for MP, 0.05 for CR, and 0.025 for HU). In the two species competition experiments, the cocultures were initiated at two initial species fractions, with the dominant species representing 99% of cells and the minority species 1% of cells. In the five species competition experiments, the cocultures were initiated at five initial species fractions, with the dominant species representing 96% of cells and the minority species each representing 1% of cells. Percent alcohol by volume (ABV) in the growth media was manipulated through the addition of pure ethanol to the sterilized juice. All competition experiments were carried out in 300 uL 96-well plates (Falcon), which were incubated at 25 °C for 24 h between each dilution step, in which 20 uL of the well mixed culture was inoculated into 180 uL of fresh ABV-corrected juice. At the end of the 7-day competition cycle, the community was diluted with phosphate-buffered saline and plated onto Wallerstein Laboratory (WL) nutrient agar, which allows for visual differentiation of the colony morphologies of the five species, and left to grow for ~72 h. The fraction and overall abundance of the species were then recorded. Due to the small volumes of our samples, we were not able to measure ethanol accumulation in the cultures, and it is therefore possible that some species combinations resulted in higher final alcohol concentrations than others, although this potential effect is minimized by the daily dilution regime.

**Growth rate measurements**. Frozen stocks of the five species were streaked out on yeast-extract peptone dextrose agar plates and left to grow at room temperature for ~48H. A single colony was then picked and transferred to a 50-mL centrifuge tube containing 5 mL of sterilized juice, adjusted to the ethanol concentration of the future growth rate measurement. Monocultures were then left to grow at room temperature for ~24H (up to 48H in the case of high ABV measurements), the OD was recorded, and the monoculture was diluted to an OD between $10^{-1}$ and $10^{-6}$ that of the overnight culture. Twenty uL of the diluted culture was inoculated into 180 uL of sterile grape juice adjusted to the same ABV as the overnight culture in 300-uL 96-well plates, and the OD at 600 nm was measured every 10 min for 300 cycles, with the plate held at 25 °C without shaking. The background OD, measured as the OD of the same volume of sterile

grape juice, was subtracted from each timepoint, and the growth rate was calculated as

$$GR = \log(OD_T / OD_{T=0})/T \qquad (1)$$

where $OD_{T=0}$ is the initial OD of the diluted culture and $OD_T$ is the OD at time $T$ (measured in hours from the initial time point). To ensure that the cultures were still in their exponential phase of growth, the growth rate was only calculated for measurements in the range of $0.01 < OD_T < 0.02$, and all growth rate estimates were based on a minimum of five measurements. This method of measuring the growth rate implicitly incorporates lag time, as strains with a longer lag times will take longer to reach a given OD than another species with the same exponential growth rate but a shorter lag time.

**Fermentation Experiments.** Fermentation experiments were carried out in 50 mL centrifuge tubes fitted with airlocks, each containing 40 mL of sterilized juice. For the monoculture fermentations, cells were inoculated at an initial density of $4 \times 10^6$ cells/mL. For the five species fermentations, cells were inoculated at one of three initial ratios: (1) all strains at $10^6$ cells/mL except for SC at $10^4$ cells/mL, (2) all strains at $10^4$ cells/mL except for MP at $4 \times 10^6$ cells/mL, and (3) all strains at $10^4$ cells/mL except for TD at $4 \times 10^6$ cells/mL. At each sampling point, three 100 uL samples were withdrawn, diluted with phosphate-buffered saline, and plated to determine the abundance of each species. In order to track the fermentation progress, an additional 1 mL sample was withdrawn, vortexed for 1 min to remove all dissolved gas, and used to measure brix using an EasyDens density and gravity meter (Anton Paar).

**Statistics and reproducibility.** All statistical analyses were performed using custom code in R. All competition outcomes were based on at least three biological replicates (see Supplementary Fig. 2). The uninterrupted fermentation experiments included three biological replicates for each initial condition in Gewürztraminer; the fermentation experiment in Merlot was not replicated. In most cases, between 100 and 500 unique colonies were counted to determine community composition, providing only limited resolution to observe low-abundance species. Accordingly, some conditions in which we observe exclusion may in fact represent coexistence with one of the species at substantially lower abundance (<1% of cells).

**Reporting summary.** Further information on research design is available in the Nature Portfolio Reporting Summary linked to this article.

## Data availability
Access to the data is publicly available through Figshare[30], including all code necessary to reproduce the figures.

## Code availability
All custom R code used in the analysis and visualization of the data is publicly available through FigShare[30].

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

## Acknowledgements
The authors would like to thank the members of the Gore lab for their constructive feedback, and the lab of Benjamin Wolfe for their comments and suggestions.

## Author contributions
S.L. and J.G. conceived of the study. S.L. carried out the experiments and analyzed the data. S.L. wrote the first draft of the paper and both authors edited it.

## Competing interests
The authors declare no competing interests.
