## [Peer Review File · Communications Biology]

Reviewers' comments:

Reviewer #1 (Remarks to the Author):

The study of wine microbiota interactions is of general interest to the scientific community due to its complexity. In recent years, the study of wine microbiota interactions has been addressed using a new methodology approach, from metagenomic to discover microbial populations, to studies of the molecular mechanisms involved in the dominance, including transcriptomic analysis.

Focusing more on competency studies, there are studies focusing on the role of commercial yeasts, including unconventional yeasts. The studies involve commercial *S. cerevisiae* strains or other commercial strains used on cocultures (*Torulasporea delbrueckii*, *Lachancea thermotolerans*, *Metschnikowia pulcherrima*, *C. zemplinina* or *Starmerella*).

One of the main problems of the present manuscript is the main objective, if the objective is the effect of alcohol on the wine microbiota the approach is not correct, in this case, the authors should analyze a more complex system including more yeast species and lactic acid bacteria. With the approach used by the authors, they are not contributing new information, is well known the effect of ethanol on *S. cerevisiae* and the majority of yeast species. Why the authors used *C. railensis*. The results of the present manuscript are not contributing to improving the knowledge of the wine community.

The second problem is the methodology used. Making the % of the yeast species by morphology is not the most recommended approach, actually, the yeast community has available a lot of molecular approach to make the yeast identification. Besides the authors only analyzed the species in the final point, why they are not analyzing the evolutions of the species along the fermentation and including the ethanol evolution?

The medium used in the competition, was sterilized juice correcting the ethanol by dilutions. With this approach, the authors modified all the medium, sugar, and nitrogen....they are included in the study other variables.

The methodology and the quality of the results don't have the quality required to be published in a journal with the scientific quality of *Communications Biology*. Besides the manuscript are not contributing with new and relevant information

Reviewer #3 (Remarks to the Author):

The manuscript, "Strong ethanol and frequency-dependent ecological interactions in a community of wine-fermenting yeasts" disentangles the effects of increasing ethanol competition and other competitive interactions in a five-species wine yeast community. The authors compare performance of community members in pairwise competitions, five-way competitions, competitions with different initial frequencies, and competitions with different amounts (or increasing amounts) of ethanol. They found that the initial community in fermenting wine or microcosms has a strong impact on competitive outcomes, but only if ethanol is not allowed to accumulate; when ethanol is allowed to accumulate, competitive outcomes become more deterministic and depend on ethanol concentration and the grape juice environment.

I found this manuscript thorough and convincing, and a useful addition to the growing body of work that treats fermented foods as model microbial communities (e.g., Wolfe et al., 2014, which is cited in the manuscript. I especially liked the idea of decoupling one important process in wine fermentation—ethanol accumulation—from other types of competitive interactions. I also appreciated how this manuscript adds to the ongoing conversation among ecologists of the roles of contingency and determinism in ecological outcomes. (In fact, I think it's a pity that the authors didn't choose to focus more on this ongoing conversation, instead keeping the focus on wine, although I respect their choice.)

I have a few, relatively minor comments. Because some of these have to do with the clarity of the manuscript and citation of previous work, I don't think the manuscript should be published before

they're addressed. But overall I enjoyed the manuscript, think the conclusions are supported by the data, and consider this a nice addition to our understanding of wine and microbial ecology.

1) There were many missing citations in the introduction and discussion sections. For example, on lines 36-38, the authors make a statement about early-succession fermenters being abundant on grape skins, but don't give a citation; the sentence that starts on line 51 with "numerous studies" should include citations to at least some of these numerous studies. This was an especially big issue in the second, third, and fourth paragraphs of the introduction and the second paragraph of the discussion, but was a general problem with the entire manuscript. I often found myself wanting to double-check a fact or definition from the manuscript, but not having a citation to use. In my opinion, this was one of the most serious issues with the manuscript (previous work must be properly credited), and must be resolved.

2) This is a bit of a semantic issue, but I always considered increasing ethanol concentrations in wine as a factor in competitive interactions, and the fact that *S. cerevisiae* is so good at producing ethanol (it is Crabtree-positive), as a competitive strategy in and of itself. The authors treat increasing ethanol concentrations as more of a quality of the environment, and imply (I don't think they mean to imply) that ethanol concentration is decoupled from biotic interactions (e.g., in line 323). I wouldn't reject the manuscript because of this minor semantic issue, but do think that the manuscript could be improved by engaging with the literature that deals with ethanol production as a competitive strategy. Some papers might include Goddard, 2008 (<https://doi.org/10.1890/07-2060.1>) and Piskur et al., 2006 (<https://doi.org/10.1016/j.tig.2006.02.002>).

3) I'm not sure the authors measured it, but I was curious about how much ethanol accumulated between transfers in experiments in which ethanol concentration is kept constant. If this information is available, it would be a very good addition to the manuscript.

4) I was confused by the statement in the discussion that frequency dependence in the yeast communities was not driven by Allee effects (line 292). My understanding of the concept of "Allee effects" is that this is simply another name for positive frequency dependence ("bistability" in the authors' language), but the authors are using the term to discuss some pretty specific mechanisms that they didn't happen to observe or don't find plausible. This could be fine, but perhaps the term "Allee effects" isn't quite the right term to use. Alternatively, I'm mistaken; in this case, an explicit definition of the term with a citation would be very helpful.

5) I'm similarly confused by the statement on lines 300-301 that observed frequency dependence stems from stronger interspecific than intraspecific competition. Wouldn't this mechanism simply lead to competitive exclusion without frequency dependence? An explanation of how exactly this mechanism results in frequency dependence (preferably with a citation, if the idea came from elsewhere!) would help reduce confusion in readers.

6) The authors discuss some diffusible molecules (pulcherrimin and killer toxins) in the discussion. It's worth noting that these molecules probably work differently in the 200 μ l experimental volumes than they do in the 40 ml wine fermentations, simply because of the difference in volume.

7) In figure 2B, it wasn't always clear to me what the criteria were to assign coexistence for HU/SC and TD/CR were here. In several of the bars, it looks like competitive exclusion. I think coexistence was assigned because of coexistence observed in Figure 1, but I'm not sure the idea is completely transferrable to these 5-species competitions (in fact, isn't that part of the point of the manuscript?). I guess I'd be satisfied if the justification were better explained.

Minor comments:

--There are many abbreviations that are first introduced in the methods, but only defined later on in the manuscript, including ABV and abbreviations for the five yeasts. The medium YPD (I know it's

standard, but the composition used should still be written out) is also not defined. These should all be defined when they first appear.

--Line 110: I was confused as to the range in OD here. Why was there variation in OD here? Wouldn't it be important to start with the same OD or number of cells for the growth rate experiments? Or is this variation to standardize to cell number?

--Line 124: "sampling point"

--Figure 4: The text references Figures 4d and 4e, but they aren't indicated on the figure. I guessed which portions of the figure these subfigures should represent, and agree that the figure should be subdivided as it is in the text.

--Figure 4: Could the statistics behind the correlations in Figure 4c-e be provided, perhaps in the supplement?

--Figure 4a: The trendline for HU seems off to me; should it be higher at ABV = 1.5 and lower at 2?

--Line 224: The authors state that growth rate declines with higher ethanol are driven by longer lag times. Could they show data supporting this, perhaps as a supplemental figure?

Line 232: Is "figure 4" meant here?

--Line 333: "led" instead of "lead"

--Line 350: Isn't this statement contradicted by the information in Figure 4C? This statement seems to better fit maximum OD, not growth rates.

--Figure 3: I was confused at first about what the shading behind the pentagrams here meant. Later on, I see that this shading was used to represent alcohol concentrations in other figures; could a legend be provided here?

We thank the three reviewers for their comments, which have helped us to strengthen our manuscript. Please find our individual responses to each comment below.

Reviewer #1 (Remarks to the Author):

The study of wine microbiota interactions is of general interest to the scientific community due to its complexity. In recent years, the study of wine microbiota interactions has been addressed using a new methodology approach, from metagenomic to discover microbial populations, to studies of the molecular mechanisms involved in the dominance, including transcriptomic analysis. Focusing more on competency studies, there are studies focusing on the role of commercial yeasts, including unconventional yeasts. The studies involve commercial *S. cerevisiae* strains or other commercial strains used on cocultures (*Torulaspota delbrueckii*, *Lachancea thermotolerans*, *Metschnikowia pulcherrima*, *C. zemplinina* or *Starmerella*). One of the main problems of the present manuscript is the main objective, if the objective is the effect of alcohol on the wine microbiota the approach is not correct, in this case, the authors should analyze a more complex system including more yeast species and lactic acid bacteria. With the approach used by the authors, they are not contributing new information, is well known the effect of ethanol on *S. cerevisiae* and the majority of yeast species. Why the authors used *C. railensis*. The results of the present manuscript are not contributing to improving the knowledge of the wine community.

While we appreciate the reviewer's concerns that our 5 species consortium does not capture the full diversity of yeasts and bacteria in an actual wine fermentation, we believe there is value in our 'bottom-up' experimental approaches which directly test the ecological interactions in the community and are necessarily limited to a tractable number of species. We are unaware of any other studies that directly map the interactions in a wine-fermenting community in this manner and strongly disagree that we do not provide any new information.

The second problem is the methodology used. Making the % of the yeast species by morphology is not the most recommended approach, actually, the yeast community has available a lot of molecular approach to make the yeast identification. Besides the authors only analyzed the species in the final point, why they are not analyzing the evolutions of the species along the fermentation and including the ethanol evolution?

We are unclear why the reviewer objects to our colony counting methods when it is a direct measurement of the community composition that is far more unbiased than most molecular approaches, such as amplicon-based ITS sequencing (see Sternes *et al*, 2017 (GigaScience) for an assessment of how molecular methods systematically overestimate the abundance of certain yeast genera). Because we are mapping the equilibrium competitive outcomes in our pairwise competitions, we chose to only measure the final community composition, but we took many samples across the time series in our uninterrupted fermentations (Figure 5).

The medium used in the competition, was sterilized juice correcting the ethanol by dilutions. With this approach, the authors modified all the medium, sugar, and nitrogen....they are included in the study other variables.

The reviewer is correct that many environmental changes occur in the wine medium during a fermentation beyond the accumulation of ethanol, but we have chosen to focus on the early stage of fermentation (up to 5% ABV) when most resources remain abundant. As we demonstrate in our experiments with artificial grape juice (Supplementary Figure 1), removing up to 75% of sugar or nitrogen has negligible impacts on the growth of our species.

The methodology and the quality of the results don't have the quality required to be published in a journal with the scientific quality of Communications Biology. Besides the manuscript are not contributing with new and relevant information

Reviewer #2 (Remarks to the Author):

In this paper the authors sought to decipher ecological interactions between wine fermentation yeast species and to identify key drivers of successional development in the yeast population by conducting binary fermentations as well as five-species mixed fermentations with different inoculation scenarios, all under conditions resembling early fermentation conditions characterized by low ethanol levels. Overall, the manuscript is scientifically and statistically sound, and the authors show that pairwise interaction studies can to some extent reliably predict interactions in complex communities. The data generated contributes valuable insights into wine fermentation population dynamics and can inform decision-making in inoculated fermentations particularly those employing *Saccharomyces* and non-*Saccharomyces* commercial starter cultures. A few minor issues need to be addressed:

1. The authors indicate that a dilution approach was used to maintain constant ethanol levels. However, given differences in the fermentation capacities of the different yeasts, no one fermentation scenario is the same. Therefore, the authors need to clearly indicate if the ethanol levels were measured first before the dilution and all adjusted to the same "constant" level and what that level actually is in %ABV.

Unfortunately we were not able to measure ethanol concentration at each dilution step due to the small volumes of the samples. Accordingly, there may have been differences in the amount of ethanol transferred at each timepoint based on how strongly fermentive the community members were. However, our experimental dilution regime (a 1/10 dilution every 24 hours) means that any accumulated ethanol over the course of a day is diluted 10x, meaning that the beginning of each growth cycle has approximately the controlled ethanol concentration. We have added clarifying text to the manuscript (lines 106-108):

"Due to the small volumes of our samples, we were not able to measure ethanol accumulation in the cultures, and it is therefore possible that some species combinations resulted in higher final alcohol concentrations than others, although this potential effect is minimized by the daily dilution."

2. Throughout the document the authors must add % before ABV when reporting the concentration.

Corrected

3. On line 97 at first mention of ABV the authors must provide the full name then (ABV)

Corrected

4. From line 89 to 112, the authors must make sure that ml is changed to mL, hours are changed H, and minutes changed to min and uL must be replaced with μ L.

Corrected

5. Throughout the document, Falcon tube(s) must be replaced with centrifuge tube(s).

Corrected

6. On line 110, 20 uL must be in words (i.e. Twenty) at the beginning of a sentence.

Corrected

7. Line 124, pint must be point

Corrected

8. Line 153, later must be latter

Corrected

9. Line 282, insert "of" after course

Corrected

10. Line 315, insert "been" after also

Corrected

11. Throughout the document, Gewurztraminer and Merlot must start with a capital letter

Corrected

Reviewer #3 (Remarks to the Author):

The manuscript, “Strong ethanol and frequency-dependent ecological interactions in a community of wine-fermenting yeasts” disentangles the effects of increasing ethanol competition and other competitive interactions in a five-species wine yeast community. The authors compare performance of community members in pairwise competitions, five-way competitions, competitions with different initial frequencies, and competitions with different amounts (or increasing amounts) of ethanol. They found that the initial community in fermenting wine or microcosms has a strong impact on competitive outcomes, but only if ethanol is not allowed to accumulate; when ethanol is allowed to accumulate, competitive outcomes become more deterministic and depend on ethanol concentration and the grape juice environment.

I found this manuscript thorough and convincing, and a useful addition to the growing body of work that treats fermented foods as model microbial communities (e.g., Wolfe et al., 2014, which is cited in the manuscript. I especially liked the idea of decoupling one important process in wine fermentation—ethanol accumulation—from other types of competitive interactions. I also appreciated how this manuscript adds to the ongoing conversation among ecologists of the roles of contingency and determinism in ecological outcomes. (In fact, I think it’s a pity that the authors didn’t choose to focus more on this ongoing conversation, instead keeping the focus on wine, although I respect their choice.)

I have a few, relatively minor comments. Because some of these have to do with the clarity of the manuscript and citation of previous work, I don’t think the manuscript should be published before they’re addressed. But overall I enjoyed the manuscript, think the conclusions are supported by the data, and consider this a nice addition to our understanding of wine and microbial ecology.

1) There were many missing citations in the introduction and discussion sections. For example, on lines 36-38, the authors make a statement about early-succession fermenters being abundant on grape skins, but don’t give a citation; the sentence that starts on line 51 with “numerous studies” should include citations to at least some of these numerous studies. This was an especially big issue in the second, third, and fourth paragraphs of the introduction and the second paragraph of the discussion, but was a general problem with the entire manuscript. I often found myself wanting to double-check a fact or definition from the manuscript, but not having a citation

to use. In my opinion, this was one of the most serious issues with the manuscript (previous work must be properly credited), and must be resolved.

We agree that there were a number of claims made in the introduction and discussion sections that should have been cited, and we have added 5 new citations while also being more careful to cite previous references at the appropriate places in the text.

2) This is a bit of a semantic issue, but I always considered increasing ethanol concentrations in wine as a factor in competitive interactions, and the fact that *S. cerevisiae* is so good at producing ethanol (it is Crabtree-positive), as a competitive strategy in and of itself. The authors treat increasing ethanol concentrations as more of a quality of the environment, and imply (I don't think they mean to imply) that ethanol concentration is decoupled from biotic interactions (e.g., in line 323). I wouldn't reject the manuscript because of this minor semantic issue, but do think that the manuscript could be improved by engaging with the literature that deals with ethanol production as a competitive strategy. Some papers might include Goddard, 2008 (<https://doi.org/10.1890/07-2060.1>) and Piskur et al., 2006 (<https://doi.org/10.1016/j.tig.2006.02.002>).

This is an excellent point, and we certainly don't mean to suggest that ethanol concentration is decoupled from competitive interactions. We have added the following text to the manuscript (lines 305-309):

“While we treat ethanol concentration in this study as an environmental factor modulating the ecological interactions between species, it is important to note the production of ethanol is itself a competitive mechanism favoring more alcohol-tolerant species. This is especially true for *Saccharomyces*, which ferments even in the presence of oxygen when respiration would be more energetically favorable (the Crabtree effect)¹³ and thereby increases its fitness through niche construction¹⁴. “

3) I'm not sure the authors measured it, but I was curious about how much ethanol accumulated between transfers in experiments in which ethanol concentration is kept constant. If this information is available, it would be a very good addition to the manuscript.

Please see our response above to Reviewer 2's first point.

4) I was confused by the statement in the discussion that frequency dependence in the yeast communities was not driven by Allee effects (line 292). My understanding of the concept of “Allee effects” is that this is simply another name for positive frequency dependence (“bistability” in the authors' language), but the authors are using the term to discuss some pretty specific mechanisms that they didn't happen to observe or don't find plausible. This could be fine, but perhaps the term “Allee effects” isn't quite the right term to use. Alternatively, I'm mistaken; in this case, an explicit definition of the term with a citation would be very helpful.

The Allee effect is a reduction in growth rate at low population density (in monoculture), which could result in a trivial form of bistability if a species is started at an initial abundance below its survival threshold. Because we do not find an Allee effect in our growth rate measurements, and because there is likely not a plausible mechanism for one to emerge, we presume the bistable outcomes are non-trivial and instead result from competition between the two species. We have added the following clarifying text to the manuscript (lines 297-298):

“Importantly, this frequency dependence is not driven by Allee effects, a reduction in growth rate at low population density which could lead the population of the species with the low initial

density to collapse. Allee effects were not observed in the mono-culture growth rate experiments for any of the five species and would be unlikely to play an important role in wine fermentation for several reasons.”

5) I’m similarly confused by the statement on lines 300-301 that observed frequency dependence stems from stronger interspecific than intraspecific competition. Wouldn’t this mechanism simply lead to competitive exclusion without frequency dependence? An explanation of how exactly this mechanism results in frequency dependence (preferably with a citation, if the idea came from elsewhere!) would help reduce confusion in readers.”

Competitive outcomes are driven by the ratios of intraspecific competition to interspecific competition, with coexistence expected when both species are primarily limited by intraspecific competition and frequency-dependence (bistability) expected when both species are primarily limited by interspecific competition. In cases where one species is primarily limited by intraspecific competition and the other is primarily limited by interspecific competition, the former is expected to exclude the latter. We have added the following clarifying text (lines 310-312):

“In the absence of Allee effects, this frequency dependence presumably stems from both species experiencing stronger interspecific (between-species) competition than intraspecific (within-species) competition”

6) The authors discuss some diffusible molecules (pulcherrimin and killer toxins) in the discussion. It’s worth noting that these molecules probably work differently in the 200 µl experimental volumes than they do in the 40 ml wine fermentations, simply because of the difference in volume.

This is a good point, and we have added the following text to the manuscript (lines 322-323):

“We also note that the effects of toxin-mediated competition may be particularly pronounced in our experiments which are characterized by high population densities in small volumes.”

7) In figure 2B, it wasn’t always clear to me what the criteria were to assign coexistence for HU/SC and TD/CR were here. In several of the bars, it looks like competitive exclusion. I think coexistence was assigned because of coexistence observed in Figure 1, but I’m not sure the idea is completely transferrable to these 5-species competitions (in fact, isn’t that part of the point of the manuscript?). I guess I’d be satisfied if the justification were better explained.

This is correct, we assigned 5-species community states based on which pairwise stable state they most resembled, which in some cases means that communities which resulted in exclusion were assigned to states with expected coexistence. We have clarified this in the text (lines 175-176):

“Each replicate was assigned to the structure it most closely resembled (which occasionally meant that replicates which resulted in exclusion were assigned to states with expected coexistence, or vice versa), and the probability of the community reaching a given structure was a function of the initial abundances.”

Minor comments:

--There are many abbreviations that are first introduced in the methods, but only defined later on in the manuscript, including ABV and abbreviations for the five yeasts. The medium YPD (I know it’s standard, but the composition used should still be written out) is also not defined. These should all be defined when they first appear.

Corrected

--Line 110: I was confused as to the range in OD here. Why was there variation in OD here? Wouldn't it be important to start with the same OD or number of cells for the growth rate experiments? Or is this variation to standardize to cell number?

While we always standardized the OD of the competing species in our coculture experiments (as described in lines 92-93), we did not standardize the OD for growth rate measurements of monocultures, which can be calculated from any initial cell density so long as it is known. We started growth rate experiments with initial dilutions over 6 orders of magnitude in order to screen for Allee effects, which we did not find.

--Line 124: "sampling point"

Corrected

--Figure 4: The text references Figures 4d and 4e, but they aren't indicated on the figure. I guessed which portions of the figure these subfigures should represent, and agree that the figure should be subdivided as it is in the text.

Thank you for catching this error in the figure, it has been corrected by adding "D" and "E" panels

--Figure 4: Could the statistics behind the correlations in Figure 4c-e be provided, perhaps in the supplement?

We have now included this information as Supplementary Table 1

--Figure 4a: The trendline for HU seems off to me; should it be higher at ABV = 1.5 and lower at 2?

The trendline was generated using a smoothed conditional mean (geom_smooth in the ggplot2 R package), and while we agree that it looks a bit off, we don't think it would be sound to manually edit it ourselves, especially when the raw data is also presented.

Line 232: Is "figure 4" meant here?

Thank you for pointing this out, we meant to reference Figure 3, specifically panel 3A. We have corrected the reference.

--Line 333: "led" instead of "lead"

Corrected

--Line 350: Isn't this statement contradicted by the information in Figure 4C? This statement seems to better fit maximum OD, not growth rates.

In saying that competitive outcomes are only weakly coupled to growth rates, we mean that although there are positive correlations they are usually not statistically significant and relatively weak (<0.7). This is also true of maximum OD (carrying capacity), and we have added that to the text as well:

"In conclusion, we show that ecological interactions play a dominant role in shaping wine-fermenting yeast communities, with monoculture growth rates and carrying capacities only weakly associated to competitive outcomes."

--Figure 3: I was confused at first about what the shading behind the pentagrams here meant. Later on, I see that this shading was used to represent alcohol concentrations in other figures; could a legend be provided here?

We have added a legend at the top of the figure.

Reviewers' comments:

Reviewer #3 (Remarks to the Author):

I appreciated the updates to the manuscript, and agree that they have improved the manuscript. Many of my comments have been addressed. However, there are some things I'm still concerned about.

While many statements in the previous manuscript have been updated with citations, there are still statements that need citations. I would have liked to have seen citations for statements given on lines 27, 40 (or is this a continuation of citation 5?), 61, 62, 65, 299, 303, 307, and 343.

I still don't agree with the authors' definition of Allee effects, but am fine with the definition they provide here, as long as it's also used elsewhere. Could they please provide a citation for this definition? If the problem is that I'm confused on the concept, surely I'm not the only potentially confused reader.

I think I don't follow the provided statistics in supplementary table 1. In the text, the authors state, "We found that both pairwise and 5-species competitive scores were always positively correlated to growth rates...except for the 5-species scores at intermediate ABV." But almost all of the correlations in the supplement weren't significant, based on the p-values in supplementary table 1 (and the few significant correlations would probably not be after p-value correction for multiple tests; the need for p-value correction could be avoided with a more sophisticated set of models). And no statistics were provided for the 5-species scores' correlation with growth rate or carrying capacity. Am I missing something? If I'm missing something, please set me straight, but it would be wise to explain in the text for readers who, like me, might be confused. It looks like you engage with this in the conclusions (in response to my earlier comments), but keep in mind that a weak correlation and a non-significant correlation aren't the same thing. This needs to be reworked.

(I do think there's a case to be made for reporting non-significant correlations in a situation like this, and there is definitely a case for doing away with p-values altogether. But this would need to be very carefully explained or justified. The standard is to report results as significant at $p < 0.05$, and even if the standard has issues moving away from it takes justification.)

I appreciated the updates to the manuscript, and agree that they have improved the manuscript. Many of my comments have been addressed. However, there are some things I'm still concerned about.

While many statements in the previous manuscript have been updated with citations, there are still statements that need citations. I would have liked to have seen citations for statements given on lines 40 (or is this a continuation of citation 5?), 61, 62, 65, 299, 303, 307, and 343.

We agree that these claims needed citations, and we thank the reviewer for reading through the manuscript so carefully. Each of these lines, as well as a few others, are now properly cited. We have added an additional five references, as well as re-citing a few previous references in the appropriate places.

I still don't agree with the authors' definition of Allee effects, but am fine with the definition they provide here, as long as it's also used elsewhere. Could they please provide a citation for this definition? If the problem is that I'm confused on the concept, surely I'm not the only potentially confused reader.

**We have now added a citation for our definition of Allee effects:
Drake, J. M. & Kramer, A. M. (2011) Allee Effects. *Nature Education Knowledge* 3(10):2**

I think I don't follow the provided statistics in supplementary table 1. In the text, the authors state, "We found that both pairwise and 5-species competitive scores were always positively correlated to growth rates...except for the 5-species scores at intermediate ABV." But almost all of the correlations in the supplement weren't significant, based on the p-values in supplementary table 1 (and the few significant correlations would probably not be after p-value correction for multiple tests; the need for p-value correction could be avoided with a more sophisticated set of models). And no statistics were provided for the 5-species scores' correlation with growth rate or carrying capacity. Am I missing something? If I'm missing something, please set me straight, but it would be wise to explain in the text for readers who, like me, might be confused. It looks like you engage with this in the conclusions (in response to my earlier comments), but keep in mind that a weak correlation and a non-significant correlation aren't the same thing. This needs to be reworked.

(I do think there's a case to be made for reporting non-significant correlations in a situation like this, and there is definitely a case for doing away with p-values altogether. But this would need to be very carefully explained or justified. The standard is to report results as significant at $p < 0.05$, and even if the standard has issues moving away from it takes justification.)

We thank the reviewer for catching our oversight in Supplementary Table 1, which now includes the 5 species correlations as well:

ABV	Pairwise Competitive Score				5 Species Competitive Score				Pairwise vs. 5SP	
	Growth Rate		K		Growth Rate		K		Corr	p
	Corr	p	Corr	p	Corr	p	Corr	p		
0	0.691	0.196	-0.048	0.939	0.660	0.225	-0.008	0.990	0.988	0.002
1	0.533	0.355	0.243	0.693	-0.161	0.796	0.833	0.080	0.644	0.241
2	0.186	0.765	0.478	0.415	0.419	0.482	0.173	0.781	0.908	0.033
3	0.272	0.657	0.486	0.406	-0.072	0.908	0.570	0.315	0.668	0.218
4	0.383	0.525	0.880	0.049	0.498	0.393	0.484	0.408	0.806	0.099
5	0.884	0.047	0.739	0.153	0.595	0.290	0.571	0.315	0.892	0.042

We agree that since the majority of our correlations are not statistically significant we do not want to overstate the relationships between these variables, but the crux of the paper is that these variables are *not* predictive of competitive success and we must therefore incorporate ecological interactions into any predictive models. Because each correlation is based on a such a small number of datapoints

(n=5), statistical significance is not easy to reach outside of very strong correlations. We merely want to point out that in almost all cases the direction of the correlation is positive, which is why we show each individual datapoint to help visualize this trend rather than providing summary statistics in the main text. We have added the following text to the manuscript to highlight that these relationships are not statistically significant (lines 229-233):

“It’s important to note that these correlations between competitive score and monoculture growth characteristics, although almost uniformly positive, rarely rise to the level of statistical significance (see **Supplementary Table 1**). While they offer some insight into why individual species are favored in a multi-species fermentation, they also highlight how they are not sufficient to predict fermentation dynamics without incorporating ecological interactions.”

REVIEWERS' COMMENTS:

Reviewer #3 (Remarks to the Author):

I really appreciated the authors' improvements and responses to my previous comments, and am very pleased with the additional citations. My one remaining comment is that the authors' treatment of the correlations in Figure 4 and supplementary table 1 (lines 218-233) is still a bit misleading. Rearranging and rephrasing some statements in this section would solve this problem.

The language around these correlations strikes me as currently too strong and not supported by the data and statistics presented. Making strong declarative statements (e.g., line 218 "both pairwise and 5-species competitive scores [were] always positively correlated to growth rates, except for the 5-species scores at intermediate ABV") without first qualifying that these correlations are non-significant, is likely to lead to confusion among readers. In line 231, the phrase "rise to statistical significance" is also confusing because a correlation either is or is not significant. I think this might have been a way for the authors to imply an important point that they mentioned in the response to reviewers--that with the small sample sizes, it's really only possible to mark general trends. If this is the case, the patterns should be caged as general trends (instead of solid results) from the get-go.

Line 220: This correlation did not become stronger as ethanol concentration increased. Instead, correlation coefficients started out high with low ethanol, went down, then went up, then down again, and finally up again. I suspect that, as the authors mentioned in the response to reviewers, a strong pattern would emerge with more replication.